# Multi-Prompt Denoised Self-Training for Open-Vocabulary Model Adaptation

## Abstract

Traditional model adaptation assumes the same vocabulary across source and target domains, which often struggles with limited transfer flexibility and efficiency while handling target domains with different vocabularies. Inspired by recent vision-language models (VLMs) that enable open-vocabulary visual recognition by reasoning on both images and texts, we study open-vocabulary model adaptation (OVMA), a new unsupervised model adaptation framework that positions a pre-trained VLM as the source model and transfers it towards arbitrary unlabelled target domains. To this end, we design a Multi-prompt denOised Self-Training (MOST) technique that exploits the synergy between vision and language to mitigate the domain discrepancies in image and text distributions simultaneously. Specifically, MOST makes use of the complementary property of multiple prompts within and across vision and language modalities, which enables joint exploitation of vision and language information and effective learning of image-text correspondences in the unlabelled target domains. Additionally, MOST captures temporal information via multi-temporal prompt learning which helps memorize previously learnt target information. Extensive experiments show that MOST outperforms the state-of-the-art consistently across 11 image recognition tasks. Codes will be released.

## 1 Introduction

Deep learning-based vision models (He et al., 2016; Dosovitskiy et al., 2020) have achieved great success in myriad image recognition tasks but at the price of laborious annotation of large-scale training images (Deng et al., 2009). To circumvent the annotation constraint, model adaptation (MA) (Liang et al., 2020; Huang et al., 2021) has been explored to transfer a vision model pre-trained in certain labelled source domains towards unlabelled target domains by mitigating the cross-domain discrepancies in image distributions. However, traditional MA (Liang et al., 2020; Huang et al., 2021; Liang et al., 2021; Xia et al., 2021; Yang et al., 2021; Ding et al., 2022; 2023) assumes that source and target domains have the same vocabulary. It struggles while handling target domains with different vocabularies, limiting its flexibility and efficiency greatly in unsupervised transfer.

Inspired by recent vision-language models (VLMs) (Radford et al., 2021) that enable open-vocabulary visual recognition by reasoning on both images and texts, we study open-vocabulary model adaptation (OVMA), a new unsupervised model adaptation (UMA) framework that positions a pre-trained VLM as the source model and transfers it towards arbitrary unlabelled target domains. OVMA requires a single pre-trained VLM only while transferring towards target domains of different vocabularies, instead of preparing multiple vocabulary-specific vision models with respective source datasets, as illustrated in Fig. 1. In addition, OVMA allows unsupervised transfer towards new domains with customized vocabulary, which greatly mitigates the image annotation constraint and facilitates deep network training while handling various new visual recognition tasks. On the other hand, the shift from traditional model adaptation toward OVMA comes with a new challenge, namely, the cross-domain discrepancies in both image distributions and text distributions.

Drawing inspiration from the recent advances in multi-prompt learning (Jiang et al., 2020; Schick & Schütze, 2020; Qin & Eisner, 2021; Yuan et al., 2021b) in natural language processing (NLP), we design Multi-prompt denOised Self-Training (MOST) that exploits the synergy between vision and language to mitigate the domain discrepancies in image and text distributions simultaneously while self-training. MOST makes use of the complementary property of multiple prompts within and

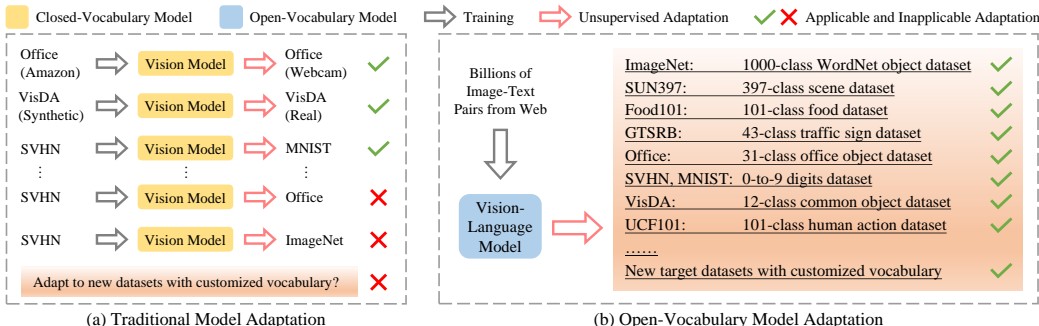

Figure 1: Traditional model adaptation transfers a vision model across datasets of the same vocabulary, which struggles while handling target datasets with different vocabularies or new datasets with customized vocabularies as illustrated in (a). Inspired by the recent open-vocabulary vision-language models (VLMs), we study open-vocabulary model adaptation, a new unsupervised model adaptation framework that positions a single pre-trained VLM as the source model and transfers it towards arbitrary unlabelled target datasets as illustrated in (b).

across vision and language modalities: it exploits VLMs to encode the image prompts (Lüddecke & Ecker, 2022; Zang et al., 2022) and text prompts (Lüddecke & Ecker, 2022; Zang et al., 2022) into an aligned vision-language feature space and fuses the encoded visual and textual features to "prompt" unsupervised self-training for denoising pseudo labels and more effective self-training and open-vocabulary model adaptation. This enables joint exploitation of vision and language information and effective learning of image-text correspondences in the unlabelled target domains. In addition, MOST captures temporal information via multi-temporal prompt learning, which helps memorize previously learnt target information by fusing the prompts encoded by the intermediate models evolved along the adaptation process.

The proposed MOST can be viewed as a new type of self-training with multi-prompt learning for the task of OVMA. It has three desirable advantages: 1) it introduces multi-visual prompt learning and multi-textual prompt learning and enables simultaneous mitigation of image and text discrepancies across domains effectively; 2) it introduces multi-temporal prompt learning along the adaptation process which allows harvesting previously learnt target information effectively; 3) it works within an aligned image-text feature space which allows multi-prompt learning not only within but also across vision, language and temporal dimensions, capturing their complementary advantages effectively.

In summary, the contributions of this work are threefold. *First*, we design a novel open-vocabulary model adaptation framework that explores multi-prompt learning upon self-training to learn effective image-text correspondences over unlabelled target images. To the best of our knowledge, this is the first work that explores multi-prompt learning for OVMA. *Second*, we design multi-prompt denoised self-training that introduces multi-prompt learning over vision, language and temporal dimensions for simultaneous mitigation of image and text discrepancies in OVMA. *Third*, extensive experiments show that the proposed multi-prompt denoised self-training outperforms the state-of-the-art consistently across multiple image recognition tasks.

## 2 RELATED WORK

**Model Adaptation** (MA), a type of unsupervised transfer learning, aims to adapt a model pre-trained on certain labelled source domains towards unlabelled target domains. Most existing MA methods can be broadly grouped into two categories. The first category employs *generative models* to compensate for the unseen source domain by generating source features (Li et al., 2020; Tian et al., 2021; Qiu et al., 2021) or images (Du et al., 2021; Yeh et al., 2021; Kurmi et al., 2021; Liu et al., 2021b). The second approach explores *self-training* that learns from unlabelled target images with predicted pseudo labels (Liang et al., 2020; Huang et al., 2021; Liang et al., 2021; Xia et al., 2021; Yang et al., 2021; Ding et al., 2022; 2023). Despite their great success, most existing methods assume the same vocabulary across the source and target domains and cannot handle target domains with different vocabulary or new domains with customized vocabulary. This limits the flexibility and efficiency of MA greatly. We study open-vocabulary model adaptation in this work, a new framework that reasons

both images and texts and allows unsupervised transfer learning towards arbitrary unlabelled target domains. We design multi-prompt denoised self-training that explores the synergy of vision and language to mitigate image and text domain gaps simultaneously in OVMA.

**Vision Language Model** (VLM) (Radford et al., 2021; Jia et al., 2021; Yuan et al., 2021a; Yu et al., 2022; Tschannen et al., 2022) aims to learn effective vision-language correlation from image-text pairs that are almost infinitely available on the Web. It has demonstrated great potential in open-vocabulary visual recognition by recognizing images with arbitrary texts. As a representative, CLIP (Radford et al., 2021) collects 400 million image-text pairs from the Web and learns rich vision-language correlation via image-text contrastive learning. Despite its great success, VLMs often suffer from degraded performance due to cross-domain discrepancies with respect to various downstream domains. Unlike recent attempts (Zhou et al., 2022b;a) that adapt VLMs by adopting prompt tuning with few-shot target images, we focus on adapting VLMs towards various downstream domains by ingeniously exploiting the unlabelled target images which are often off-the-shelf available in abundance.

**Multi-Prompt Learning** explores complementary advantages of different prompts (Jiang et al., 2020) which was originally designed for effective transfer of large language models in NLP. Most existing methods can be broadly grouped into three categories. The first is *prompt ensembling* that creates multiple unanswered prompts for an input to predict via uniform averaging (Jiang et al., 2020; Schick & Schütze, 2020; Yuan et al., 2021b), weighted averaging (Jiang et al., 2020; Qin & Eisner, 2021; Schick & Schütze, 2020), majority voting (Lester et al., 2021; Hambardzumyan et al., 2021), etc. The second exploits *prompt augmentation* that provides several answered prompts for an input for better predictions, where most studies focus on the selection (Gao et al., 2020; Lu et al., 2021; Liu et al., 2021a) and ordering (Lu et al., 2021; Kumar & Talukdar, 2021; Guu et al., 2018) of answered prompts. The third works by *prompt composition or decomposition* (Han et al., 2022; Cui et al., 2021), which constructs multiple sub-prompts for better predictions.

## 3 METHOD

### 3.1 PRELIMINARIES OF VISION-LANGUAGE MODEL

**Vision-language model (VLM) training.** VLM (Radford et al., 2021; Jia et al., 2021; Yuan et al., 2021a; Yu et al., 2022; Tschannen et al., 2022) learns effective vision-language correlation from image-text pairs that are almost infinitely available on the Web (Radford et al., 2021; Schuhmann et al., 2021). The training involves a VLM $F = \{F^I, F^T\}$ where $F^I$ and $F^T$ denote an image encoder and a text encoder respectively, and an image-text dataset $D_s = \{(x_n^I, x_n^T)\}_{n=1}^N$ where $x_n^I$ and $x_n^T$ stand for an image sample and its paired text sample. Given $F$ and $D_s$, rich vision-language correlation can be learnt with a vision-language training objective such as image-text contrast (Radford et al., 2021) as follows:

$$\mathcal{L}_{\text{VLM}} = -\sum_{i=1}^N \log \frac{\exp\left(z_i^I \cdot z_i^T / \tau\right)}{\sum_{j=1}^N \exp(z_i^I \cdot z_j^T / \tau)} - \sum_{i=1}^N \log \frac{\exp\left(z_i^T \cdot z_i^I / \tau\right)}{\sum_{j=1}^N \exp(z_i^T \cdot z_j^I / \tau)}, \tag{1}$$

where the two terms on the right denote image-to-text and text-to-image contrastive losses respectively. The notations $z_i^I = F^I(x_i^I)$ and $z_i^T = F^T(x_i^T)$ stand for the encoded image and text features respectively, $\tau$ denotes a temperature parameter (Wu et al., 2018), and "·" stands for the inner-product that measures the cosine similarity between two features.

**VLM inference.** A pre-trained VLM can perform open-vocabulary image recognition on arbitrary unlabelled target domains by reasoning on both images and texts (Radford et al., 2021). Given an arbitrary unlabelled target dataset $D = \{X^I, X^T\}$, $X^I = \{x_n^I\}_{n=1}^N$ stands for $N$ unlabelled images and $X^T = \{x_m^T\}_{m=1}^M$ denotes $M$ class names of interest, e.g., $X^T = \{\text{car, bus, ..., bike, person}\}$. The pre-trained VLM predicts the probability of an image $x^I$ belonging to class $x^T$ by:

$$p_{x^I \to x^T} = z^I \cdot z^T, \tag{2}$$

where $z^I = F^I(x^I)$, $z^T = F^T(x^T)$. Theoretically, VLMs can work with any class names $X^T$ and thus achieve open-vocabulary image recognition. Note $X^T = \{x_m^T\}_{m=1}^M$ contains $M$ target-domain class names but provides no information of which image belongs to which class name (Radford et al., 2021).

**Domain discrepancies leads to degraded performance.** VLMs often suffer from degraded performance due to cross-domain discrepancies with respect to various target domains (Li et al., 2022). For example, for domain discrepancies in text distributions, VLMs are largely pre-trained on the source domains that consist of free-form sentences while the target domains generally provide only raw class names, where such discrepancies between source and target domains often lead to degraded performance. For domain discrepancies in image distributions, VLMs are largely pre-trained on normal images from the internet while most target datasets have quite different domains, e.g., images in synthetic, Clipart, Sketch styles etc., where such discrepancies usually lead to degraded performance. Previous works (Radford et al., 2021; Zhou et al., 2022b; Li et al., 2022; Bahng et al., 2022) also show that there are little overlap between VLM training data and test target data, and properly tackle the gaps between them via text or visual prompt learning or model finetuning could improve the performance on target datasets.

### 3.2 Definition of Open-vocabulary Model Adaptation (OVMA)

This work focuses on the task of OVMA, a new unsupervised model adaptation (UMA) framework that transfers a pre-trained VLM $F = \{F^I, F^T\}$ towards an arbitrary unlabelled target domain $D = \{X^I, X^T\}$ with certain unsupervised training losses, i.e., $\mathcal{L}_{\text{OVMA}} = \mathcal{L}_{\text{unsupervised}}(X^I, X^T; F^I, F^T)$. Take self-training (Zhu, 2005; Zou et al., 2018) as an example. Given $X^I = \{x_n^I\}_{n=1}^N$ and $X^T = \{x_m^T\}_{m=1}^M$, the unsupervised training loss on unlabelled target data can be formulated as the following:

$$\hat{y}_n^I = \arg\max_m \; z_n^I \cdot z_m^T, \qquad \mathcal{L}_{\text{ST}} = -\sum_{n=1}^N \log \frac{\sum_{m=1}^M \exp\left(z_n^I \cdot z_m^T/\tau\right) \times \mathbb{1}(\hat{y}_n^I == m)}{\sum_{m=1}^M \exp(z_n^I \cdot z_m^T/\tau)}, \quad (3)$$

where $z_n^I$ and $z_m^T$ denote the encoded image and text features, i.e., $z_n^I = F^I(x_n^I)$ and $z_m^T = F^T(x_m^T)$. $\hat{y}_n^I$ stands for the pseudo label of $x_n^I$.

Note the unsupervised training is often unstable and susceptible to collapse if we optimize VLM image encoder and text encoder concurrently (Li et al., 2022). Hence, we freeze the VLM text encoder during unsupervised model adaptation for stable adaptation.

### 3.3 Multi-Prompt Denoised Self-training

We tackle the challenge of OVMA from a perspective of multi-prompt learning (Jiang et al., 2020; Schick & Schütze, 2020). As illustrated in Fig. 2, we design Multi-prompt denOised Self-Training (MOST) that introduces multi-visual prompt learning and multi-textual prompt learning over self-training to mitigate the domain discrepancies in image and text distributions simultaneously.

In addition, MOST captures temporal information via multi-temporal prompt learning, which helps memorize previously learnt target information by fusing the prompts encoded by the intermediate models evolved along the adaptation process.

**Multi-textual prompt learning** fuses the text features generated from different text prompts, aiming to leverage the complementary information of multiple text prompts (i.e., various text descriptions for a class (Lüddecke & Ecker, 2022; Zang et al., 2022)) to mitigate the cross-domain discrepancy in text distributions. It employs a Large Language Model (Brown et al., 2020; Wang & Komatsuzaki, 2021) (LLM) to generate multiple text prompts for a given class name and then encodes them by the VLM text encoder. The encoded text features are then fused in a two-step manner: 1) uniformly average the multiple text features to acquire an initial prompt centroid 2) calculate the final prompt centroid by weighted average where the weight of each feature is the distance between it and the initial prompt centroid. This two-step operation allows smooth prompt fusion by weighting down the effect of corner cases, which is important for multi-textual prompt learning as the LLM-generated prompts are not always reliable (e.g., when experiencing generation failures, LLM may generate only a full stop character "." or a random word).

Given a class name $x_m^T \in X^T$, we employ the Large Language Model (Brown et al., 2020) to generate $K$ text prompts $\{x_{(m,1)}^T, x_{(m,2)}^T, ..., x_{(m,K)}^T\}$ and then the VLM text encoder $F^T$ to encode the generated prompts to acquire text features $\{z_{(m,1)}^T, z_{(m,2)}^T, ..., z_{(m,K)}^T\}$ (i.e., $z_{(m,k)}^T = F^T(x_{(m,k)}^T)$).

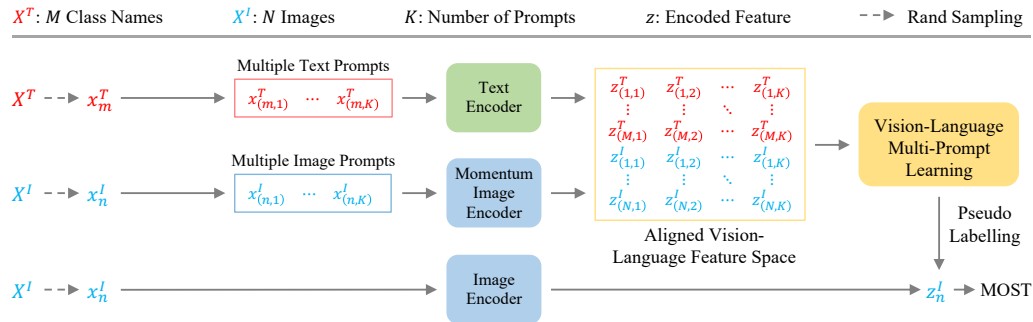

Figure 2: **Overview of multi-prompt denoised self-training (MOST).** MOST exploits the complementary property of multiple prompts within and across vision and language modalities, which enables joint exploitation of vision and language information and effective learning of image-text correspondences in the unlabelled target domains. Besides, MOST captures temporal information via multi-temporal prompt learning along the training process, which helps memorize previously learnt target information.

The text features are then fused in a two-step manner to get the final text prompt centroid $\delta_m^T$:

$$\delta_m^{T_{\text{initial}}} = \frac{1}{K} \sum_{k=1}^{K} z_{(m,k)}^T, \quad \delta_m^T = \sum_{k=1}^{K} (z_{(m,k)}^T \cdot \delta_m^{T_{\text{initial}}}) \times z_{(m,k)}^T, \tag{4}$$

where "$\cdot$" denotes inner-product and $(z_{(m,k)}^T \cdot \delta_m^{T_{\text{initial}}})$ measures the distance between $z_{(m,k)}^T$ and $\delta_m^{T_{\text{initial}}}$.

**Multi-visual prompt learning** fuses the image features generated from multiple image prompts, aiming to utilize the complementary property of multiple image prompts (i.e., various image descriptions for a class (Lüddecke & Ecker, 2022; Zang et al., 2022)) for mitigating the cross-domain discrepancy in image distributions. Given an image, it employs certain off-the-shelf image augmentation policies (Cubuk et al., 2020) to generate multiple image prompts, encodes them with the VLM image encoder, and fuses the encoded image features in a class-wise manner. Since target images are unlabelled, we generated pseudo labels for class-wise image feature fusion. The class-wise feature fusion allows category-wise image prompt consolidation, which is crucial to multi-visual prompt learning due to the abundance of target images and the encoded image features. In addition, it simplifies vision-language multi-prompt learning greatly (described in the later paragraphs) as multi-textual prompt learning also works in a category-wise manner. Besides, with multi-temporal prompt learning (described in the later paragraphs), it allows to dynamically select image prompts using pseudo labels along the adaptation process to describe each class visually.

Given an image $x_n^I \in X^I$, we adopt the off-the-shelf image augmentation policies in (Cubuk et al., 2020) to generate $K$ image prompts $\{x_{(n,1)}^I, x_{(n,2)}^I, ..., x_{(n,K)}^I\}$ and then the VLM image encoder $F^I$ to encode the generated image prompts to acquire image features $\{z_{(n,1)}^I, z_{(n,2)}^I, ..., z_{(n,K)}^I\}$ (i.e., $z_{(n,k)}^I = F^I(x_{(n,k)}^I)$). Finally, the encoded features are fused in a class-wise manner to get the image prompt centroid $\delta_m^I$:

$$\delta_m^I = \frac{1}{\sum_n^N \sum_{k=1}^K \mathbb{1}(\hat{y}_{(n,k)}^I == m)} \sum_n^N \sum_{k}^K z_{(n,k)}^I \times \mathbb{1}(\hat{y}_{(n,k)}^I == m), \tag{5}$$

where $\mathbb{1}(\hat{y}_{(n,k)}^I == m)$ returns "1" if $\hat{y}_{(n,k)}^I = m$ else 0. Note $\hat{y}_{(n,k)}^I = \arg\max_m z_{(n,k)}^I \cdot z_m^T$ denotes the pseudo label of $x_{(n,k)}^I$. Note we employ the momentum update of $F^I$ in the vision prompt fusion for stable feature encoding and better capturing of temporal information, as shown in Fig. 2.

**Temporal vision-language multi-prompt learning** exploits the synergy between vision and language by fusing multiple text prompts and multiple image prompts over an aligned vision-language feature space. It employs the text and image prompt centroids as starting point and updates them with the image prompt centroids generated by the intermediate VLM image encoder evolved along the

adaptation process. This enables multi-prompt learning not only within but also across vision and language modalities, capturing the complementary advantages of vision and language information effectively. In addition, the updating also achieves **multi-temporal prompt learning** that captures previously learnt target information effectively. Note we conduct temporal fusion for image prompts only as the VLM text encoder is frozen during the adaptation process.

Specifically, we use the text and image prompt centroids $\delta_m^T$ and $\delta_m^I$ to initialize the image-text prompt centroid $\delta_m^{IT}$ and keep updating $\delta_m^{IT}$ with $\delta_m^I$ along the adaptation process as follows:

$$\delta_m^{IT_{\text{initial}}} = \delta_m^I + \delta_m^T, \quad \delta_m^{IT*} \leftarrow \lambda \delta_m^{IT} + (1 - \lambda)\delta_m^I, \tag{6}$$

where $\delta_m^{IT}$ and $\delta_m^{IT*}$ denote the image-text prompt centroid before and after one update, respectively. $\lambda$ is a coefficient that controls the update speed of temporal fusion. Note the first part denotes *vision-text prompt fusion* while the second part denotes *temporal prompt fusion*.

**Multi-prompt denoised self-training.** Given image-text prompt centroid $\delta_m^{IT}$, target images, $X^I = \{x_n^I\}_{n=1}^N$ and target class names $X^T = \{x_m^T\}_{m=1}^M$, we employ $\delta_m^{IT}$ to "prompt" unsupervised self-training, which can be formulated as follows:

$$\tilde{y}_n^I = \arg\max_m \ (z_n^I \cdot z_m^T) \times (z_n^I \cdot \delta_m^{IT}), \tag{7}$$

$$\mathcal{L}_{\text{MOST}} = -\sum_{n=1}^N \log \frac{\sum_{m=1}^M \exp(z_n^I \cdot z_m^T / \tau) \times \mathbb{1}(\tilde{y}_n^I == m)}{\sum_{m=1}^M \exp(z_n^I \cdot z_m^T / \tau)}, \tag{8}$$

where $z_n^I$ and $z_m^T$ denote the encoded image and text features, i.e., $z_n^I = F^I(x_n^I)$ and $z_m^T = F^T(x_m^T)$. $\tilde{y}_n^I$ stands for the pseudo label of $x_n^I$ generated with $\delta_m^{IT}$. The image-text prompt centroid $\delta_m^{IT}$ captures rich target image and text information. It is thus more invariant to visual and textual domain discrepancies and can "prompt" self-training to generate more accurate pseudo labels.

## 4 EXPERIMENTS

This section presents experiments including benchmarking over 11 widely adopted image recognition datasets, spanning multi-domain datasets with object images captured from several domains (e.g., synthetic, sketch and clipart domains) to single-domain datasets for some specific visual tasks (e.g., the recognition of foods, traffic signs, natural textures and human actions). Due to the space limit, more details about the datasets and implementation details are provided in the appendix.

### 4.1 MOST ON MULTI-DOMAIN DATASETS

Tables 1-3 report the image classification results on 4 representative multi-domain datasets. The experiments were conducted with 3 representative backbones, i.e., ResNet-50, ResNet-101 and ViT-B/16. It can be seen that our MOST achieves superior performance consistently over various domains as compared with state-of-the-art methods. Besides, MOST outperforms CLIP substantially on Office (S)ynthetic domain, Office-Home (C)lipart domain and Adaptiope (S)ynthetic domain with 15.6%, 10.4% and 13.9% accuracy improvement, respectively, showing that MOST can well handle the target domains with large domain discrepancies, i.e., Synthetic and Clipart styles.

### 4.2 MOST ON SINGLE-DOMAIN DATASETS

Table 4 reports the image classification over 5 popular single-domain datasets. The experiments were conducted with 3 representative backbones, i.e., ResNet-50, ResNet-101 and ViT-B/16 (the results with ResNet-101 are provided in the appendix). We can observe that MOST outperforms the state-of-the-arts by large margins consistently over different task-specific datasets, demonstrating that it can effectively handle various new visual recognition tasks by using unlabelled data. In addition, MOST brings substantial improvements upon CLIP over SUN397 (e.g., +11.0% on ViT-B/16) and GTSRB (e.g., +16.8% on ViT-B/16), showing that MOST can well tackle new image classification tasks with very specific objectives, e.g., indoor/outdoor scene and German traffic sign recognition.

Table 1: OVMA performance on multi-domain datasets of Office, Office-Home and Adaptiope.

| ViT-B/16 | Office | | | | | Office-Home | | | | | Adaptiope | | | |
|---|---|---|---|---|---|---|---|---|---|---|---|---|---|---|
| | A | W | D | S | Mean | A | C | P | R | Mean | P | R | S | Mean |
| CLIP (Radford et al., 2021) | 77.9 | 79.4 | 76.9 | 56.7 | 72.7 | 74.4 | 58.5 | 79.6 | 79.4 | 72.9 | 82.6 | 78.2 | 45.9 | 68.9 |
| ST (Zhu, 2005) | 78.6 | 81.1 | 78.3 | 68.6 | 76.6 | 77.8 | 62.5 | 81.3 | 80.3 | 75.4 | 86.7 | 82.0 | 49.5 | 72.7 |
| CBST (Zou et al., 2018) | 79.1 | 80.7 | 78.5 | 68.9 | 76.8 | 77.3 | 62.8 | 81.7 | 80.7 | 75.6 | 86.9 | 83.2 | 50.1 | 73.4 |
| CRST (Zou et al., 2019) | 78.8 | 81.2 | 79.1 | 69.0 | 77.0 | 78.1 | 63.1 | 81.4 | 81.1 | 75.9 | 87.1 | 83.9 | 50.7 | 73.9 |
| SHOT (Liang et al., 2020) | 79.2 | 81.1 | 81.2 | 67.1 | 77.1 | 77.9 | 64.3 | 80.9 | 81.5 | 76.1 | 88.3 | 84.7 | 51.2 | 74.7 |
| MUST (Li et al., 2022) | 79.0 | 81.4 | 79.5 | 69.2 | 77.2 | 77.7 | 63.9 | 82.1 | 81.4 | 76.2 | 88.8 | 85.3 | 51.5 | 75.2 |
| MOST (Ours) | **84.3** | **82.8** | **81.3** | **72.3** | **80.1** | **78.9** | **68.9** | **85.7** | **82.4** | **78.9** | **91.8** | **88.1** | **59.8** | **79.9** |

| ResNet-50 | Office | | | | | Office-Home | | | | | Adaptiope | | | |
|---|---|---|---|---|---|---|---|---|---|---|---|---|---|---|
| | A | W | D | S | Mean | A | C | P | R | Mean | P | R | S | Mean |
| CLIP (Radford et al., 2021) | 72.9 | 68.9 | 73.1 | 48.2 | 65.7 | 64.6 | 42.1 | 71.9 | 71.9 | 62.6 | 74.5 | 66.2 | 35.8 | 58.8 |
| ST (Zhu, 2005) | 75.2 | 66.8 | 71.3 | 44.1 | 64.3 | 66.7 | 38.6 | 72.0 | 73.8 | 62.7 | 75.7 | 70.7 | 26.7 | 57.7 |
| CBST (Zou et al., 2018) | 75.2 | 67.8 | 72.2 | 51.1 | 66.5 | 68.1 | 41.5 | 73.6 | 74.5 | 64.4 | 77.2 | 71.1 | 34.3 | 60.8 |
| CRST (Zou et al., 2019) | 76.4 | 67.4 | 74.5 | 52.3 | 67.6 | 68.3 | 42.3 | 74.8 | 75.3 | 65.1 | 78.3 | 71.2 | 36.2 | 61.9 |
| SHOT (Liang et al., 2020) | 77.5 | 70.1 | 76.8 | 54.8 | 69.8 | 68.4 | 44.2 | 75.7 | 75.6 | 65.9 | 78.5 | 72.4 | 36.8 | 62.5 |
| MOST (Ours) | **79.6** | **75.3** | **80.3** | **55.0** | **72.5** | **68.6** | **47.9** | **78.2** | **77.4** | **68.0** | **80.7** | **75.6** | **37.8** | **64.7** |

| ResNet-101 | Office | | | | | Office-Home | | | | | Adaptiope | | | |
|---|---|---|---|---|---|---|---|---|---|---|---|---|---|---|
| | A | W | D | S | Mean | A | C | P | R | Mean | P | R | S | Mean |
| CLIP (Radford et al., 2021) | 73.2 | 73.8 | 75.1 | 50.2 | 68.0 | 69.5 | 47.8 | 74.3 | 74.2 | 66.4 | 75.9 | 69.0 | 35.3 | 60.0 |
| ST (Zhu, 2005) | 74.4 | 74.2 | 73.8 | 54.3 | 69.1 | 71.4 | 43.2 | 74.9 | 75.0 | 66.1 | 78.4 | 71.8 | 37.8 | 62.6 |
| CBST (Zou et al., 2018) | 74.6 | 75.9 | 72.9 | 58.1 | 70.3 | 72.3 | 44.9 | 77.7 | 76.2 | 67.7 | 79.5 | 73.3 | 41.5 | 64.7 |
| CRST (Zou et al., 2019) | 75.3 | 76.6 | 73.4 | 58.5 | 70.9 | 73.4 | 45.9 | 78.4 | 76.8 | 68.6 | 80.1 | 75.2 | 43.7 | 66.3 |
| SHOT (Liang et al., 2020) | 76.9 | 78.2 | 75.1 | 59.0 | 72.3 | 73.5 | 47.2 | 79.1 | 77.4 | 69.3 | 81.9 | 76.3 | 44.1 | 67.4 |
| MOST (Ours) | **80.1** | **81.2** | **77.5** | **61.9** | **75.1** | **74.6** | **51.2** | **82.6** | **78.9** | **71.8** | **85.3** | **78.8** | **45.7** | **69.9** |

Table 2: OVMA performance on large-scale multi-domain dataset VisDA.

| **VisDA Synthesis Domain** | | | | | | | | | | | | | |
|---|---|---|---|---|---|---|---|---|---|---|---|---|---|
| ViT-B/16 | plane | bcycl | bus | car | horse | knife | mcycl | person | plant | sktbrd | train | truck | Per-class |
| CLIP (Radford et al., 2021) | 98.5 | 99.7 | 64.6 | 92.5 | 99.7 | 96.8 | 85.3 | 98.4 | 99.8 | 79.4 | 66.4 | 73.4 | 87.8 |
| ST (Zhu, 2005) | 97.2 | **99.9** | 60.4 | 84.5 | 99.8 | 98.6 | 92.5 | 99.7 | **99.9** | 79.3 | 74.2 | 84.4 | 89.2 |
| CBST (Zou et al., 2018) | 98.4 | 99.7 | 67.3 | 85.2 | 99.8 | 99.1 | 95.3 | **99.9** | 99.4 | 83.4 | 83.4 | 87.4 | 91.5 |
| CRST (Zou et al., 2019) | 98.1 | 98.2 | 70.5 | 86.5 | 99.8 | 98.7 | 94.3 | 99.8 | 86.7 | 88.7 | 86.1 | 91.9 |
| SHOT (Liang et al., 2020) | 99.6 | 99.1 | 74.6 | 86.3 | 98.3 | **99.3** | **96.4** | 96.1 | 99.7 | 87.5 | 90.1 | 87.3 | 92.2 |
| MUST (Li et al., 2022) | 98.7 | 99.2 | 76.3 | 86.4 | 99.6 | 99.2 | 95.3 | 99.3 | 99.8 | 89.2 | 89.9 | 82.6 | 92.9 |
| MOST (Ours) | **99.7** | 99.7 | **78.9** | **86.6** | **99.9** | **99.3** | **96.4** | 99.4 | 99.8 | **91.9** | **90.8** | **93.2** | **94.6** |

| **VisDA Real Domain** | | | | | | | | | | | | | |
|---|---|---|---|---|---|---|---|---|---|---|---|---|---|
| ViT-B/16 | plane | bcycl | bus | car | horse | knife | mcycl | person | plant | sktbrd | train | truck | Per-class |
| CLIP (Radford et al., 2021) | 98.9 | 91.0 | 90.5 | 65.7 | 98.6 | 89.1 | 95.3 | 56.5 | 90.2 | 96.8 | 93.8 | **75.8** | 86.8 |
| ST (Zhu, 2005) | **99.4** | 87.3 | 92.5 | 68.3 | 98.1 | 90.4 | 94.6 | 69.3 | 91.2 | 96.7 | 94.5 | 66.4 | 87.3 |
| CBST (Zou et al., 2018) | 99.3 | 89.2 | 91.3 | **76.9** | 98.2 | 89.5 | 95.4 | 68.1 | 88.4 | 96.4 | 94.1 | 64.2 | 87.5 |
| CRST (Zou et al., 2019) | 99.1 | 90.7 | 91.4 | 64.5 | 99.1 | 93.4 | 95.1 | 68.2 | 91.3 | 96.8 | 95.3 | 67.2 | 87.6 |
| SHOT (Liang et al., 2020) | 99.3 | 92.8 | 91.9 | 65.3 | 98.7 | 95.2 | 94.5 | 67.7 | 92.1 | 96.9 | 95.4 | 67.9 | 88.1 |
| MUST (Li et al., 2022) | 99.2 | 95.7 | **92.6** | 56.9 | 99.1 | **98.6** | 96.0 | 67.0 | **93.5** | **98.8** | **96.9** | 68.1 | 88.5 |
| MOST (Ours) | 99.2 | **95.9** | 92.1 | 66.1 | **99.2** | 97.8 | **96.7** | **70.8** | 92.7 | 98.4 | 96.2 | 74.6 | **90.0** |

Table 3: OVMA performance on multi-domain datasets of DomainNet.

| Method | **ViT-B/16** | | | | | | | **ResNet-50** | | | | | | |
|---|---|---|---|---|---|---|---|---|---|---|---|---|---|---|
| | Clipart | Info | Paint | Quick | Real | Sketch | Mean | Clipart | Info | Paint | Quick | Real | Sketch | Mean |
| CLIP (Radford et al., 2021) | 69.7 | 47.8 | 65.0 | 14.5 | 82.0 | 62.4 | 56.9 | 51.9 | 39.1 | 52.1 | 6.4 | 74.7 | 47.4 | 45.3 |
| ST (Zhu, 2005) | 72.5 | 51.3 | 68.7 | 12.4 | 83.7 | 64.3 | 58.8 | 55.4 | 40.5 | 54.8 | 4.3 | 76.2 | 48.3 | 46.5 |
| CBST (Zou et al., 2018) | 74.3 | 56.8 | 69.8 | 13.4 | 83.1 | 67.1 | 60.7 | 56.3 | 40.7 | 56.2 | 5.6 | 77.4 | 48.1 | 47.3 |
| CRST (Zou et al., 2019) | 75.6 | 56.9 | 71.3 | 14.8 | 83.3 | 68.2 | 61.7 | 57.9 | 41.8 | 57.1 | 6.2 | 78.2 | 49.5 | 48.4 |
| SHOT (Liang et al., 2020) | 75.9 | 57.4 | 71.5 | 15.1 | 83.3 | 68.8 | 62.0 | 60.3 | 45.8 | 60.5 | 5.1 | 78.9 | 54.1 | 50.8 |
| MUST (Li et al., 2022) | 76.1 | 57.5 | 71.6 | 14.2 | 84.4 | 68.9 | 62.1 | - | - | - | - | - | - | - |
| MOST (Ours) | **77.6** | **59.0** | **73.1** | **18.2** | **86.1** | **70.1** | **64.0** | **62.7** | **47.2** | **61.3** | **7.2** | **80.2** | **54.4** | **52.2** |

## 4.3 OVMA ON GENERAL DATASET IMAGENET

Table 5 presents the image classification results on ImageNet. It can be seen that MOST achieves superior performance as compared with state-of-the-art unsupervised methods, demonstrating the effectiveness of MOST over the very diverse and large-scale ImageNet. Besides, MOST surpasses 16-

Table 4: OVMA performance on single-domain datasets of various image recognition tasks.

| Method | ViT-B | | | | | | ResNet-50 | | | | | |
|---|---|---|---|---|---|---|---|---|---|---|---|---|
| | SUN397 | Food101 | GTSRB | DTD | UCF101 | Mean | SUN397 | Food101 | GTSRB | DTD | UCF101 | Mean |
| CLIP (Radford et al., 2021) | 60.8 | 85.6 | 32.5 | 44.5 | 64.1 | 57.5 | 54.0 | 73.1 | 25.0 | 39.8 | 56.0 | 49.5 |
| ST (Zhu, 2005) | 65.8 | 88.2 | 32.8 | 45.0 | 67.0 | 59.7 | 59.0 | 74.4 | 20.5 | 35.8 | 56.4 | 49.2 |
| CBST (Zou et al., 2018) | 63.2 | 89.5 | 37.6 | 44.3 | 68.1 | 60.5 | 63.7 | 78.2 | 27.4 | 38.7 | 59.5 | 53.5 |
| CRST (Zou et al., 2019) | 64.7 | 89.1 | 39.7 | 45.3 | 68.6 | 61.4 | 64.2 | 76.5 | 30.1 | 39.4 | 61.3 | 54.3 |
| SHOT (Liang et al., 2020) | 66.1 | 89.6 | 41.2 | 46.3 | 69.4 | 62.5 | 65.1 | 77.3 | 34.6 | 41.2 | 62.7 | 56.1 |
| MUST (Li et al., 2022) | 67.7 | 89.4 | 42.7 | 46.5 | 70.6 | 63.3 | - | - | - | - | - | - |
| MOST (Ours) | **71.8** | **91.1** | **49.3** | **52.7** | **73.9** | **67.7** | **65.7** | **79.5** | **39.6** | **49.4** | **65.6** | **59.9** |

shot supervised methods by a clear margin (i.e., +7.2%), validating its advantages as a unsupervised method to mitigate the image annotation constraint and facilitate deep network training while handling new visual recognition tasks.

Table 5: Comparison with few-shot supervised adaptation methods and unsupervised adaption methods on ImageNet. All methods use the same CLIP ViT-B/16 model.

| Method | CLIP | Supervised with 16 Labels per Class | | Unsupervised | | |
|---|---|---|---|---|---|---|
| | | CoCoOp (Zhou et al., 2022a) | CoOp (Zhou et al., 2022b) | ST (Zhu, 2005) | MUST (Li et al., 2022) | MOST (Ours) |
| ImageNet Accuracy | 68.3 | 71.0 | 71.5 | 76.5 | 77.7 | **78.7** |

## 4.4 DISCUSSION

**Generalization across different domains and tasks.** We examine the generalization of MOST with respect to image recognition tasks and domains. Specifically, we perform extensive evaluations over 10 widely studied multi-domain (Saenko et al., 2010; Venkateswara et al., 2017; Ringwald & Stiefelhagen, 2021; Peng et al., 2017) and single-domain (Deng et al., 2009; Xiao et al., 2010; Bossard et al., 2014; Stallkamp et al., 2011; Cimpoi et al., 2014; Soomro et al., 2012) datasets as described in Table 9. Experimental results in Tables 1- 5 show that the proposed MOST achieves superior image recognition performance consistently across different domains and tasks.

**Generalization across different backbones.** We study the generalization of MOST by assessing it with three popular image recognition backbones, including two CNNs (i.e., ResNet-50 and ResNet-101) and one Transformer (i.e., ViT-B/16). Results in Tables 1- 5 and the tables in appendix B show that our MOST works effectively and consistently over different image recognition backbones.

Table 6: Ablation studies of MOST with ViT-B/16 on Office dataset.

| Method | Vision-Language Multi-Prompt Learning | | Multi-Temporal Prompt Learning | Office (Mean) |
|---|---|---|---|---|
| | Multi-Visual Prompt Learning | Multi-Textual Prompt Learning | | |
| CLIP | | | | 72.7 |
| ST | | | | 76.6 |
| | ✓ | | | 77.5 |
| | | ✓ | | 78.2 |
| | ✓ | ✓ | | 78.7 |
| MOST | ✓ | ✓ | ✓ | **80.1** |

**Ablation study.** We conduct ablation studies with ViT-B/16 on Office as shown in Table 6. As the core of the proposed MOST, we examine how our designed *multi-visual prompt learning* , *multi-textual prompt learning* and *multi-temporal prompt learning* contribute to the overall performance of open-vocabulary model adaptation. As shown in Table 6, including either multi-visual prompt learning or multi-textual prompt learning above self-training improves performance clearly, showing that image and text prompts fusion help mitigate cross-domain discrepancies in image distributions and text distributions and can "prompt" unsupervised self-training with more accurate pseudo label prediction. In addition, combining multi-visual and multi-textual prompt learning performs clearly better, indicating that the two types of multi-prompt learning complement each other by working from orthogonal vision and language perspectives. Furthermore, including *multi-temporal prompt learning* upon vision-language multi-prompt learning (MOST in the last row) performs the best. It demonstrates the importance of multi-temporal prompt learning that helps memorize previously learnt target information along the training process.

Figure 3: Pseudo label accuracy along the unsupervised adaptation process in MOST (with ViT-B/16).

**Parameter study.** The parameter $\lambda$ in Eq. 6 controls the update speed of temporal fusion. We investigate $\lambda$ by varying it from 0.9 to 0.9999 progressively, as shown in Table 7. It can be seen that varying $\lambda$ does not affect MOST clearly. The performance drops a bit while $\lambda = 0.9$, largely because a fast update may lead to unstable multi-temporal prompt learning that only captures local information within each training batch.

Table 7: Parameter ablations with ViT-B/16 on Office. The default is marked in gray .

| Parameter $\lambda$ | 0.9 | 0.99 | 0.999 | 0.9999 |
|---|---|---|---|---|
| Office (Mean) | 79.6 | **80.1** | 80.1 | 80.0 |

**Comparison with multi-prompt learning methods.** We compare MOST with multi-prompt learning strategies that explore complementary advantages of different prompts via uniform averaging (Jiang et al., 2020; Schick & Schütze, 2020; Yuan et al., 2021b), weighted averaging (Jiang et al., 2020; Qin & Eisner, 2021; Schick & Schütze, 2020), majority voting (Lester et al., 2021; Hambardzumyan et al., 2021). As Table 8 shows, existing

Table 8: Comparison with other multi-prompt learning methods with ViT-B/16 on Office.

| Method | Office (Mean) |
|---|---|
| ST + Uniform Averaging (Jiang et al., 2020) | 77.2 |
| ST + Weighted Averaging (Qin & Eisner, 2021) | 77.4 |
| ST + Majority Voting (Lester et al., 2021) | 77.1 |
| MOST (Ours) | **80.1** |

multi-prompt learning methods do not perform well, largely because they were designed for NLP without considering the joint exploitation of vision and language modalities and the information memorization during unsupervised transfer. MOST instead learns and memorizes effective image-text correspondences in the unlabelled target domains via joint exploitation of vision and language information, which are essential to MOST.

**Pseudo label accuracy.** Fig. 3 shows the pseudo label accuracy along the unsupervised adaptation process. MOST generates much more accurate pseudo labels than the vanilla self-training (ST) and the state-of-the-art MUST. The superior pseudo label accuracy is largely attributed to the proposed multi-prompt learning which helps capture rich target image and text information that is more invariant to visual and textual domain discrepancies and can "prompt" better unsupervised self-training.

**Comparison with MUST.** MUST (Li et al., 2022) tackles unsupervised adaptation of VLMs from the perspective of Masked Image Modelling (He et al., 2021) that heavily relies on Transformer backbones (Dosovitskiy et al., 2020). As a comparison, the proposed MOST works from the perspective of multiple-prompt learning that is independent to vision backbones. Thus, MOST can seamlessly work on different vision backbones like CNNs and Transformers as shown in Tables 1-4. In addition, Tables 1-5 show that MOST outperforms MUST clearly, largely because MUST exploits vision information largely while MOST exploits both vision and language information jointly which is better aligned with the objective of MOST.

Due to the space limit, we provide more dataset details, experiments and discussions in the appendix.

## 5    CONCLUSION

This paper presents MOST, a novel open-vocabulary model adaptation framework that explores multi-prompt learning to learn effective image-text correspondences over unlabelled target images. MOST exploits multi-prompt learning over vision, language and temporal dimensions for simultaneous mitigation of image and text discrepancies across domains. It requires merely a pre-trained VLM but achieves effective and efficient UMA towards arbitrary unlabelled target domains, demonstrating its superiority in facilitating deep network training while handling arbitrary new visual recognition tasks and domains. Extensive experiments show that MOST achieves superb recognition performance consistently across different backbones and image recognition tasks and domains. Moving forward, we will explore multi-prompt learning for other vision tasks such as image generation.

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

## A  APPENDIX

### A.1  DATASET DETAILS

We benchmark our proposed MOST extensively over 11 widely adopted image recognition datasets. As Table 9 shows, the 11 datasets have rich diversity, spanning multi-domain datasets with object images captured from several domains (e.g., real-world, synthetic, art, product and clipart domains) to single-domain datasets with real-world images for some specific visual task (e.g., the recognition of common objects, indoor and outdoor scenes, foods, traffic signs, natural textures and human actions). Below please find the detail of each dataset.

**Office** (Saenko et al., 2010) includes 31-class images collected from Amazon (A), Webcam (W) and DSLR (D) domains which have 2817, 795 and 498 images, respectively. In addition to the original three domains in Office dataset, we further include an office Synthetic (S) domain for benchmarking our MOST comprehensively. The Synthetic (S) domain is provided by (Ringwald & Stiefelhagen, 2021) and consists of 3100 images.

**Office-home** (Venkateswara et al., 2017) consists of 65-class images collected from Art (A), Clipart (C), Product (P) and Real-World (R) domains which include 2496, 4464, 4503 and 4450 images, respectively.

**Adaptiope** (Ringwald & Stiefelhagen, 2021) has 123-class images collected from 3 domains, i.e., Product (P), Real-World (R) and Synthetic (S), where each domain has 12300 images.

**VisDA** (Peng et al., 2017) has over 280K images of 12 classes from Synthetic (S) domain and Real-World (R) domain, which contain 152397 and 127760 images, respectively.

**DomainNet** (Peng et al., 2019) includes 345-class images from Clipart, Infograph, Painting, Quick-Draw, Real-World and Sketch domains which include 48129, 51605, 72266, 172500, 172947 and 69128 images, respectively.

**ImageNet** (Deng et al., 2009) includes about 1.2M images that are uniformly distributed across the one thousand categories. The category annotation of ImageNet follows WordNet hierarchy and every image is annotated with one category label.

**SUN397** (Xiao et al., 2010) has been proposed for scene recognition, which contains 39700 images covering 397 well-sampled scene categories, including indoor scenes and outdoor scenes.

**Food101** (Bossard et al., 2014) is a real-world food dish dataset for fine-grained image recognition. The dataset consists of 101K images that cover 101 classes. Specifically, each class includes 250 cleaned test images and 750 purposely uncleaned training images.

**GTSRB** (Stallkamp et al., 2011) is a real-world dataset for traffic signs recognition, which includes 50K images collected from various street scenes in Germany. These images have been labelled into 43 categories, including a training subset with 39209 images and a testing subset with 12630 images.

**Describable Textures (DTD)** (Cimpoi et al., 2014) is a collection of textural images for texture recognition. This dataset consists of 5640 images with 47 categories, which have been uniformly separated into training, validation, and test subsets, where each subset contains 40 images per class. For each image, a main category and a list of the joint attributes are provided.

**UCF101** (Soomro et al., 2012) has been proposed for benchmarking human action recognition with videos. It includes about 13K video clips of 101 actions, which are collected from YouTube. The video clips in the dataset have a resolution of 320x240 pixels and a frame rate of 25 FPS.

Table 9: Image recognition datasets used for open-vocabulary model adaptation benchmark.

| Dataset | Classes | Images | Domains | Description |
|---|---|---|---|---|
| Office (Saenko et al., 2010) | 31 | 4,110 | 4 | Office objects from Amazon, DSLR, Webcam and Synthetic domains. |
| Office-home (Venkateswara et al., 2017) | 65 | 15,588 | 4 | Office and Home objects from Art, Clipart, Product and Real-World domains. |
| Adaptiope (Ringwald & Stiefelhagen, 2021) | 123 | 36,900 | 3 | Class-balanced object dataset with Product, Real-Life and Synthetic domains. |
| VisDA (Peng et al., 2017) | 12 | 207,785 | 2 | A large-scale common object dataset with synthetic and real domains. |
| DomainNet (Peng et al., 2019) | 345 | 586,575 | 6 | Common objects from Clipart, Infograph, Painting, Quickdraw, Real and Sketch domains. |
| ImageNet (Deng et al., 2009) | 1,000 | 1,281,167 | 1 | A large-scale real-world object dataset with a wide range of categories. |
| SUN397 (Xiao et al., 2010) | 397 | 76,129 | 1 | A real-world indoor and outdoor scenes dataset for scene understanding. |
| Food101 (Bossard et al., 2014) | 101 | 75,750 | 1 | A real-world food dish dataset for food recognition. |
| GTSRB (Stallkamp et al., 2011) | 43 | 26,640 | 1 | A real-world german traffic sign dataset for sign recognition. |
| DTD (Cimpoi et al., 2014) | 47 | 3,760 | 1 | A real-world describable texture image dataset for texture perception. |
| UCF101 (Soomro et al., 2012) | 101 | 9,537 | 1 | A real-world human action video dataset for action recognition. |

## A.2 IMPLEMENTATION DETAILS

We conduct experiments with three popular backbones, i.e., ResNet-50 (He et al., 2016), ResNet-101 (He et al., 2016) and ViT-B (Dosovitskiy et al., 2020) pre-trained by CLIP (Radford et al., 2021). In training, we employ AdamW optimizer (Loshchilov & Hutter, 2017) with a weight decay of $0.05$, and set the initial learning rate as $1e-5$ which is adjusted with a cosine learning rate schedule. We use $2$ GPUs with batch size $64$ and the unsupervised adaptation training adds only a small amount of computation overhead after VLM pre-training. We set input image size as $224 \times 224$ and employ data augmentation policies of "RandomResizedCrop+Flip+RandAug" (Cubuk et al., 2020) to generate multiple image prompts. The momentum VLM image encoder is updated with a momentum coefficient of $0.99$. All results except on ImageNet are obtained with above implementation details. For the large-scale ImageNet, we follow the implementations in (Li et al., 2022) and use 16 GPUs with batch size 1024. During evaluation, we simply use the center-cropped image.

## A.3 EXPERIMENTS WITH DIFFERENT BACKBONES

In Section 4.6 in the main manuscript, we study the generalization of our proposed MOST by assessing it with three popular image recognition backbones, including two CNNs (i.e., ResNet-50 and ResNet-101) and one Transformer (i.e., ViT-B/16). Table 2 in the main manuscript provides the full results of the three backbones on multi-domain datasets Office, Office-Home and Adaptiope. Due to the space limit, Tables 3 and 4 in the main manuscript only provide partial results for VisDA and other 5 single-domain datasets.

Here we provide the full result versions of the Table 3 and Table 4 in the main manuscript, as shown in Table 10 and Table 11, which further demonstrate that our MOST works effectively and consistently over different image recognition backbones.

## A.4 PSEUDO CODES OF MULTI-PROMPT DENOISED SELF-TRAINING

We provide the pseudo codes of our proposed multi-prompt denoised self-training, as shown in Algorithm 1. Note Algorithm 1 describes the unsupervised adaptation process in a epoch-wise manner for simple illustration and presentation. In experiments, we implement Algorithm 1 in a iteration-wise manner with mini-batches. Besides, Lines 7-8 in Algorithm 1 can be skipped in the first training iteration as the model has not been updated at that time.

Note, in traditional multi-prompt learning, the prompts are updated while the model is generally fixed. Differently, our MOST introduces multi-prompt learning into self-training, where the prompts and the model are alternatively updated as illustrated in Line 8 and Line 10 in Algorithm 1. In this way, MOST captures temporal information via multi-temporal prompt learning, which helps

Table 10: OVMA performance (with three widely adopted backbone networks) on large-scale multi-domain dataset VisDA.

| ViT-B/16 | VisDA Synthesis Domain | | | | | | | | | | | | |
|---|---|---|---|---|---|---|---|---|---|---|---|---|---|
| | plane | bcycl | bus | car | horse | knife | mcycl | person | plant | sktbrd | train | truck | Per-class |
| CLIP (Radford et al., 2021) | 98.5 | 99.7 | 64.6 | 92.5 | 99.7 | 96.8 | 85.3 | 98.4 | 99.8 | 79.4 | 66.4 | 73.4 | 87.8 |
| ST (Zhu, 2005) | 97.2 | 99.9 | 60.4 | 84.5 | 99.8 | 98.6 | 92.5 | 99.7 | 99.9 | 79.3 | 74.2 | 84.4 | 89.2 |
| CBST (Zou et al., 2018) | 98.4 | 99.7 | 67.3 | 85.2 | 99.8 | 99.1 | 95.3 | 99.9 | 99.4 | 83.4 | 83.4 | 87.4 | 91.5 |
| CRST (Zou et al., 2019) | 98.1 | 98.2 | 70.5 | 86.5 | 98.6 | 98.7 | 94.3 | 98.8 | 97.8 | 86.7 | 88.7 | 86.1 | 91.9 |
| SHOT (Liang et al., 2020) | 99.6 | 99.1 | 74.6 | 86.3 | 98.3 | 99.3 | 96.4 | 96.1 | 99.7 | 87.5 | 90.1 | 87.3 | 92.2 |
| MUST (Li et al., 2022) | 98.7 | 99.2 | 76.3 | 86.4 | 99.6 | 99.2 | 95.3 | 99.3 | 99.8 | 89.2 | 89.9 | 82.6 | 92.9 |
| MOST (Ours) | 99.7 | 99.7 | 78.9 | 86.6 | 99.9 | 99.3 | 96.4 | 99.4 | 99.8 | 91.9 | 90.8 | 93.2 | 94.6 |

| ViT-B/16 | VisDA Real Domain | | | | | | | | | | | | |
|---|---|---|---|---|---|---|---|---|---|---|---|---|---|
| | plane | bcycl | bus | car | horse | knife | mcycl | person | plant | sktbrd | train | truck | Per-class |
| CLIP (Radford et al., 2021) | 98.9 | 91.0 | 90.5 | 65.7 | 98.6 | 89.1 | 95.3 | 56.5 | 90.2 | 96.8 | 93.8 | 75.8 | 86.8 |
| ST (Zhu, 2005) | 99.4 | 87.3 | 92.5 | 68.3 | 98.1 | 90.4 | 94.6 | 69.3 | 91.2 | 96.7 | 94.5 | 66.4 | 87.3 |
| CBST (Zou et al., 2018) | 99.3 | 89.2 | 91.3 | 76.9 | 98.2 | 89.5 | 95.4 | 68.1 | 88.4 | 96.4 | 94.1 | 64.2 | 87.5 |
| CRST (Zou et al., 2019) | 99.1 | 90.7 | 91.4 | 64.5 | 99.1 | 93.4 | 95.1 | 68.2 | 91.3 | 96.8 | 95.3 | 67.2 | 87.6 |
| SHOT (Liang et al., 2020) | 99.3 | 92.8 | 91.9 | 65.3 | 98.7 | 95.2 | 94.5 | 67.7 | 92.1 | 96.9 | 95.4 | 67.9 | 88.1 |
| MUST (Li et al., 2022) | 99.2 | 95.7 | 92.6 | 56.9 | 99.1 | 98.6 | 96.0 | 67.0 | 93.5 | 98.8 | 96.9 | 68.1 | 88.5 |
| MOST (Ours) | 99.2 | 95.9 | 92.1 | 66.1 | 99.2 | 97.8 | 96.7 | 70.8 | 92.7 | 98.4 | 96.2 | 74.6 | 90.0 |

| ResNet-50 | VisDA Synthesis Domain | | | | | | | | | | | | |
|---|---|---|---|---|---|---|---|---|---|---|---|---|---|
| | plane | bcycl | bus | car | horse | knife | mcycl | person | plant | sktbrd | train | truck | Per-class |
| CLIP (Radford et al., 2021) | 96.0 | 99.1 | 43.4 | 92.4 | 98.5 | 94.5 | 69.6 | 92.1 | 99.1 | 46.6 | 53.0 | 41.5 | 77.1 |
| ST (Zhu, 2005) | 94.2 | 99.3 | 38.9 | 75.2 | 97.4 | 93.7 | 78.5 | 94.6 | 99.3 | 63.4 | 57.8 | 88.2 | 81.7 |
| CBST (Zou et al., 2018) | 95.7 | 99.6 | 37.2 | 73.3 | 96.8 | 95.6 | 84.5 | 96.8 | 99.2 | 68.7 | 59.2 | 89.4 | 83.1 |
| CRST (Zou et al., 2019) | 96.6 | 99.9 | 30.1 | 71.3 | 99.9 | 99.1 | 92.8 | 99.9 | 99.4 | 75.0 | 61.1 | 97.2 | 85.1 |
| SHOT (Liang et al., 2020) | 97.3 | 99.9 | 43.7 | 73.4 | 98.6 | 98.6 | 91.9 | 99.7 | 99.1 | 77.3 | 68.9 | 84.4 | 86.0 |
| MOST (Ours) | 97.6 | 99.8 | 57.2 | 84.7 | 99.9 | 98.7 | 91.7 | 99.8 | 100 | 79.2 | 74.5 | 83.1 | 88.8 |

| ResNet-50 | VisDA Real Domain | | | | | | | | | | | | |
|---|---|---|---|---|---|---|---|---|---|---|---|---|---|
| | plane | bcycl | bus | car | horse | knife | mcycl | person | plant | sktbrd | train | truck | Per-class |
| CLIP (Radford et al., 2021) | 97.3 | 82.1 | 83.0 | 55.4 | 96.7 | 73.4 | 91.1 | 59.9 | 86.6 | 93.4 | 91.8 | 73.8 | 82.0 |
| ST (Zhu, 2005) | 97.6 | 78.1 | 99.7 | 65.9 | 96.2 | 79.3 | 90.1 | 62.8 | 82.9 | 94.2 | 89.1 | 74.3 | 84.1 |
| CBST (Zou et al., 2018) | 95.8 | 83.2 | 80.3 | 54.5 | 96.8 | 92.2 | 92.1 | 74.8 | 91.6 | 88.8 | 89.8 | 76.0 | 84.9 |
| CRST (Zou et al., 2019) | 96.9 | 86.9 | 83.1 | 71.1 | 93.4 | 91.9 | 91.7 | 80.3 | 90.2 | 89.4 | 88.5 | 65.6 | 85.7 |
| SHOT (Liang et al., 2020) | 96.5 | 85.4 | 85.4 | 59.6 | 96.3 | 94.8 | 92.7 | 80.3 | 92.4 | 90.5 | 90.4 | 75.4 | 86.6 |
| MOST (Ours) | 97.2 | 87.2 | 88.2 | 78.1 | 97.2 | 95.1 | 93.0 | 81.5 | 92.1 | 91.2 | 92.7 | 65.6 | 88.2 |

| ResNet-101 | VisDA Synthesis Domain | | | | | | | | | | | | |
|---|---|---|---|---|---|---|---|---|---|---|---|---|---|
| | plane | bcycl | bus | car | horse | knife | mcycl | person | plant | sktbrd | train | truck | Per-class |
| CLIP (Radford et al., 2021) | 96.8 | 99.4 | 24.2 | 87.5 | 98.9 | 96.7 | 83.1 | 58.2 | 99.3 | 61.2 | 47.1 | 72.4 | 77.0 |
| ST (Zhu, 2005) | 95.2 | 99.6 | 26.7 | 84.3 | 99.1 | 97.2 | 84.2 | 91.3 | 99.5 | 68.4 | 57.6 | 81.2 | 82.0 |
| CBST (Zou et al., 2018) | 96.7 | 99.8 | 27.3 | 74.5 | 99.9 | 99.5 | 93.8 | 99.9 | 100 | 73.1 | 62.3 | 97.0 | 85.3 |
| CRST (Zou et al., 2019) | 96.9 | 99.9 | 42.0 | 78.6 | 99.9 | 98.9 | 93.5 | 99.9 | 99.9 | 73.0 | 72.0 | 94.4 | 87.4 |
| SHOT (Liang et al., 2020) | 98.5 | 99.7 | 39.9 | 83.1 | 100 | 98.5 | 97.8 | 99.1 | 100 | 79.3 | 81.7 | 91.3 | 89.0 |
| MOST (Ours) | 97.8 | 99.8 | 47.5 | 85.5 | 100 | 98.8 | 96.6 | 99.9 | 100 | 81.1 | 83.2 | 92.2 | 90.2 |

| ResNet-101 | VisDA Real Domain | | | | | | | | | | | | |
|---|---|---|---|---|---|---|---|---|---|---|---|---|---|
| | plane | bcycl | bus | car | horse | knife | mcycl | person | plant | sktbrd | train | truck | Per-class |
| CLIP (Radford et al., 2021) | 97.8 | 83.7 | 87.9 | 76.2 | 97.4 | 77.9 | 93.8 | 53.7 | 84.3 | 90.7 | 91.0 | 67.2 | 83.4 |
| ST (Zhu, 2005) | 97.4 | 84.7 | 86.6 | 75.2 | 97.1 | 80.5 | 94.1 | 69.7 | 89.6 | 91.1 | 92.3 | 68.7 | 85.5 |
| CBST (Zou et al., 2018) | 97.3 | 86.5 | 87.7 | 70.6 | 97.3 | 93.8 | 93.3 | 74.5 | 91.7 | 89.1 | 91.5 | 69.1 | 86.8 |
| CRST (Zou et al., 2019) | 97.5 | 82.9 | 86.3 | 82.2 | 97.8 | 93.1 | 95.4 | 68.5 | 94.4 | 91.3 | 93.2 | 66.8 | 87.4 |
| SHOT (Liang et al., 2020) | 97.3 | 88.6 | 88.6 | 69.8 | 97.3 | 94.2 | 92.9 | 80.4 | 91.8 | 92.7 | 92.3 | 69.2 | 87.9 |
| MOST (Ours) | 97.8 | 89.1 | 88.3 | 78.3 | 97.3 | 94.5 | 94.7 | 82.1 | 92.8 | 93.6 | 93.8 | 69.5 | 89.3 |

memorize previously learnt target information by fusing the prompts encoded by the intermediate models evolved along the adaptation process.

Table 11: OVMA performance (with three widely adopted backbone networks) on single-domain datasets of various image recognition tasks.

| Method | ViT-B/16 | | | | | | ResNet-50 | | | | | |
|---|---|---|---|---|---|---|---|---|---|---|---|---|
| | SUN397 | Food101 | GTSRB | DTD | UCF101 | Mean | SUN397 | Food101 | GTSRB | DTD | UCF101 | Mean |
| CLIP (Radford et al., 2021) | 60.8 | 85.6 | 32.5 | 44.5 | 64.1 | 57.5 | 54.0 | 73.1 | 25.0 | 39.8 | 56.0 | 49.5 |
| ST (Zhu, 2005) | 65.8 | 88.2 | 32.8 | 45.0 | 67.0 | 59.7 | 59.0 | 74.4 | 20.5 | 35.8 | 56.4 | 49.2 |
| CBST (Zou et al., 2018) | 63.2 | 89.5 | 37.6 | 44.3 | 68.1 | 60.5 | 63.7 | 78.2 | 27.4 | 38.7 | 59.5 | 53.5 |
| CRST (Zou et al., 2019) | 64.7 | 89.1 | 39.7 | 45.3 | 68.6 | 61.4 | 64.2 | 76.5 | 30.1 | 39.4 | 61.3 | 54.3 |
| SHOT (Liang et al., 2020) | 66.1 | 89.6 | 41.2 | 46.3 | 69.4 | 62.5 | 65.1 | 77.3 | 34.6 | 41.2 | 62.7 | 56.1 |
| MUST (Li et al., 2022) | 67.7 | 89.4 | 42.7 | 46.5 | 70.6 | 63.3 | - | - | - | - | - | - |
| MOST (Ours) | 71.8 | 91.1 | 49.3 | 52.7 | 73.9 | 67.7 | 65.7 | 79.5 | 39.6 | 49.4 | 65.6 | 59.9 |

| Method | ResNet-101 | | | | | |
|---|---|---|---|---|---|---|
| | SUN397 | Food101 | GTSRB | DTD | UCF101 | Mean |
| CLIP (Radford et al., 2021) | 51.5 | 82.3 | 27.5 | 37.8 | 58.3 | 51.4 |
| ST (Zhu, 2005) | 56.5 | 79.9 | 23.6 | 35.4 | 60.2 | 51.1 |
| CBST (Zou et al., 2018) | 65.7 | 81.5 | 28.3 | 37.3 | 60.5 | 54.6 |
| CRST (Zou et al., 2019) | 61.4 | 80.7 | 31.4 | 37.3 | 63.0 | 54.7 |
| SHOT (Liang et al., 2020) | 63.7 | 81.4 | 33.9 | 42.5 | 64.3 | 57.1 |
| MUST (Li et al., 2022) | - | - | - | - | - | - |
| MOST (Ours) | 67.5 | 83.4 | 38.2 | 48.1 | 66.2 | 60.6 |

Table 12: OVMA performance (with three widely adopted backbone networks) on multi-domain datasets of DomainNet.

| Method | ViT-B/16 | | | | | | | ResNet-50 | | | | | | |
|---|---|---|---|---|---|---|---|---|---|---|---|---|---|---|
| | Clipart | Info | Paint | Quick | Real | Sketch | Mean | Clipart | Info | Paint | Quick | Real | Sketch | Mean |
| CLIP (Radford et al., 2021) | 69.7 | 47.8 | 65.0 | 14.5 | 82.0 | 62.4 | 56.9 | 51.9 | 39.1 | 52.1 | 6.4 | 74.7 | 47.4 | 45.3 |
| ST (Zhu, 2005) | 72.5 | 51.3 | 68.7 | 12.4 | 83.7 | 64.3 | 58.8 | 55.4 | 40.5 | 54.8 | 4.3 | 76.2 | 48.3 | 46.5 |
| CBST (Zou et al., 2018) | 74.3 | 56.8 | 69.8 | 13.4 | 83.1 | 67.1 | 60.7 | 56.3 | 40.7 | 56.2 | 5.6 | 77.4 | 48.1 | 47.3 |
| CRST (Zou et al., 2019) | 75.6 | 56.9 | 71.3 | 14.8 | 83.3 | 68.2 | 61.7 | 57.9 | 41.8 | 57.1 | 6.2 | 78.2 | 49.5 | 48.4 |
| SHOT (Liang et al., 2020) | 75.9 | 57.4 | 71.5 | 15.1 | 83.3 | 68.8 | 62.0 | 60.3 | 45.8 | 60.5 | 5.1 | 78.9 | 54.1 | 50.8 |
| MUST (Li et al., 2022) | 76.1 | 57.5 | 71.6 | 14.2 | 84.4 | 68.9 | 62.1 | - | - | - | - | - | - | - |
| MOST (Ours) | 77.6 | 59.0 | 73.1 | 18.2 | 86.1 | 70.1 | 64.0 | 62.7 | 47.2 | 61.3 | 7.2 | 80.2 | 54.4 | 52.2 |

| Method | ResNet-101 | | | | | | |
|---|---|---|---|---|---|---|---|
| | Clipart | Info | Paint | Quick | Real | Sketch | Mean |
| CLIP (Radford et al., 2021) | 58.8 | 41.5 | 58.0 | 8.9 | 77.4 | 53.8 | 49.8 |
| ST (Zhu, 2005) | 61.4 | 47.5 | 61.7 | 6.1 | 78.9 | 55.2 | 51.8 |
| CBST (Zou et al., 2018) | 63.2 | 48.3 | 62.5 | 6.7 | 79.4 | 56.1 | 52.7 |
| CRST (Zou et al., 2019) | 64.3 | 49.4 | 63.2 | 6.9 | 80.2 | 57.8 | 53.6 |
| SHOT (Liang et al., 2020) | 66.4 | 49.4 | 65.4 | 7.9 | 80.8 | 59.2 | 54.9 |
| MUST (Li et al., 2022) | - | - | - | - | - | - | - |
| MOST (Ours) | 69.6 | 50.8 | 65.9 | 9.5 | 82.5 | 60.4 | 56.4 |

---

**Algorithm 1** Multi-Prompt Denoised Self-training.

---

**Require:** Target images $X^I$, target class names $X^T$ and a pre-trained vision-language model $F = \{F^I, F^T\}$
**Ensure:** Adapted vision-language model $F$
1: **Initialization:**
2: Calculate text prompt centroid $\delta_m^T$ using $X^T$ and $F$ via Eq. 4
3: Calculate image prompt centroid $\delta_m^I$ using $X^I$ and $F$ via Eq. 5
4: Initialize image-text prompt centroid $\delta_m^{IT}$ using $\delta_m^T$ and $\delta_m^I$ as in the left part of Eq. 6
5: **for** $epoch = 1$ **to** $Max\_Epoch$ **do**
6:     **Pseudo Label Generation:**
7:     Calculate new image prompt centroid $\delta_m^I$ using $X^I$ and the updated $F$ using Eq. 5
8:     Update image-text prompt centroid $\delta_m^{IT}$ with new image prompt centroid $\delta_m^I$ as in the right part of Eq. 6
9:     Generate pseudo labels $Y^I$ with the updated image-text prompt centroid $\delta_m^{IT}$ via Eq. 7
10:     **Network Optimization with Pseudo Labels:**
11:     Optimize $F$ using pseudo labels $Y^I$ via Eq. 8
12: **end for**
13: **return** $F$

---

### A.5 HOW LLM-GENERATED TEXT PROMPTS AFFECT OTHER METHODS

As described in Section 3 and discussed in Section 4.4, our proposed MOST adopts GPT-3 (Brown et al., 2020) as the large language model (LLM) to generate multiple text prompts for a given class for mitigating cross-domain discrepancy in text distributions. For comprehensively benchmarking MOST, we provide the results of the state-of-the-art methods using the same LLM-generated text prompts as those used in MOST. Table 13 presents the results on dataset Office with backbone ViT-B/16.

We can observe that directly using LLM-generated text prompts for these methods improves the performance slightly. Beside, it can be seen that our MOST still outperforms the state-of-the-arts that used LLM-generated text prompts, largely because MOST conducts multi-prompt learning that captures, fuses and updates the prompts generated by LLM and utilize them to denoise pseudo labels.

Table 13: Results of the state-of-the-art methods with the text prompts generated from Large Language Models (Brown et al., 2020).

| ViT-B/16 | Office | | | | |
|---|---|---|---|---|---|
| | A | W | D | S | Mean |
| ST (Zhu, 2005) | 78.6 | 81.1 | 78.3 | 68.6 | 76.6 |
| ST (Zhu, 2005) + LLM (Brown et al., 2020) | 79.2 | 82.0 | 78.9 | 70.1 | 77.5 |
| CBST (Zou et al., 2018) | 79.1 | 80.7 | 78.5 | 68.9 | 76.8 |
| CBST (Zou et al., 2018) + LLM (Brown et al., 2020) | 80.1 | 81.4 | 79.3 | 70.3 | 77.7 |
| CRST (Zou et al., 2019) | 78.8 | 81.2 | 79.1 | 69.0 | 77.0 |
| CRST (Zou et al., 2019) + LLM (Brown et al., 2020) | 79.1 | 82.1 | 80.3 | 70.2 | 77.9 |
| SHOT (Liang et al., 2020) | 79.2 | 81.1 | 81.2 | 67.1 | 77.1 |
| SHOT (Liang et al., 2020) + LLM (Brown et al., 2020) | 80.7 | 81.9 | 81.7 | 68.9 | 78.3 |
| MUST (Li et al., 2022) | 79.0 | 81.4 | 79.5 | 69.2 | 77.2 |
| MUST (Li et al., 2022) + LLM (Brown et al., 2020) | 81.2 | 82.1 | 80.7 | 70.2 | 78.5 |
| MOST (Ours) | **84.3** | **82.8** | **81.3** | **72.3** | **80.1** |

## A.6 RELATIONS TO OPEN-SET, CLASS-INCREMENTAL AND PARTIAL DOMAIN ADAPTATION

Different from traditional domain adaptation that assumes the same vocabulary across source and target domains, this work studies open-vocabulary model adaptation (OVMA), a new unsupervised model adaptation (UMA) framework that positions a pre-trained VLM as the source model and transfers it towards arbitrary unlabelled target domains.

We note that there are several other domain adaptation frameworks which also aim to handle the situation where the source and target domains have different vocabularies. In this section, we briefly introduce their frameworks and clarify the difference between them and the studied OVMA.

Specifically, open-set domain adaptation (Panareda Busto & Gall, 2017; Saito et al., 2018; Liu et al., 2019), class-incremental domain adaptation (Kundu et al., 2020; Xu et al., 2021) and partial domain adaptation (Cao et al., 2018; 2019; Zhang et al., 2018), are proposed to handle the situation where the source and target domains have different vocabularies. However, all these frameworks have certain limitations as compared the studied OVMA.

For example, **open-set domain adaptation** (Panareda Busto & Gall, 2017; Saito et al., 2018; Liu et al., 2019) adds an extra class called "unknown" to both source and target domains such that it allows open-set adaptation by treating all the classes that are not shared between source and target domains as the "unknown" class. However, open-set domain adaptation can merely classify all new target classes/concepts as a single "unknown" class even in an ideal case, which fails to respectively recognize new target classes/concepts, limiting its flexibility and efficiency greatly in unsupervised transfer. Differently, OVMA allows to respectively recognize arbitrary new target categories/concepts, which is much more flexible.

**Class-incremental domain adaptation** (Kundu et al., 2020; Xu et al., 2021) integrates domain adaptation and class-incremental learning (using one-shot or few-shot labelled target images) such that it allows to recognize new target classes/concepts during domain adaptation. However, it generally requires one-shot or few-shot labelled target images for each new class as a prerequisite, while OVMA is unsupervised and can work for new classes without requiring labelled target images.

**Partial domain adaptation** (Cao et al., 2018; 2019; Zhang et al., 2018) assumes that the label set of target domain is a subset of the label set of source domain. Differently, the studied OVMA does not have this constraint as it can work with arbitrary target classes (Radford et al., 2015).

## A.7 MULTI-PROMPT LEARNING MITIGATES CROSS-DOMAIN DISCREPANCIES

Prompts have been widely explored in transfer learning in NLP, which can help reformulate the downstream tasks (i.e., target domains) such that they look more like those learnt during pre-training (i.e., source domain) (Liu et al., 2023). In this way, multi-prompt learning can mitigate the discrepancies between the training domain (i.e., source domain) and the testing domain (i.e., target domain) by providing suitable descriptions for the target-domain tasks with respect to the model pre-trained on the source domain, ultimately leading to improved performance on target domains.

In vision-language models (VLMs) that handle multiple data modalities, both images and texts can be used as the prompts to describe a given class (Lüddecke & Ecker, 2022; Zang et al., 2022), where better class concept descriptions often improve the downstream image classification tasks (Lüddecke & Ecker, 2022; Zang et al., 2022). For example, we could use both the text of "a four-wheeled road vehicle that is powered by an engine" and a car image to describe the concept of class "car" (Lüddecke & Ecker, 2022; Zang et al., 2022). In this work, we focus on open-vocabulary model adaptation and introduce multi-prompt leaning to select and fuse multiple image and text prompts, which aims to find and create suitable descriptions for each target-domain class with respect to the VLM pre-trained with source domain (i.e., the web-scale image-text pair dataset (Radford et al., 2021)), ultimately improving the classification performance on the target domain.

## A.8 MOST WITH DIFFERENT LLMS

As described in the main manuscript, our proposed MOST employs GPT-3 (Brown et al., 2020) as the large language model (LLM) to generate multiple text prompts for a given class. Specifically, for all datasets, we query the large language model with the following input:

"Describe what a/an `[class name]`, a type of `[dataset name]`, looks like."

In this section, we study how the adoption of LLM affects MOST by implementing MOST with different LLMs, including GPT-3 (Brown et al., 2020), GPT-2 (Radford et al., 2019) and GPT-J-6B (Wang & Komatsuzaki, 2021). Experimental results in Table 14 show that the change of LLM does not affect MOST clearly, demonstrating that MOST can work effectively and consistently with different qualities of text prompts (generated by different LLMs).

Table 14: MOST with different large language models. Experiments are conducted with ViT-B/16 on dataset Office. The default implementation is highlighted in gray .

| Method | Office (Mean) | Office-home (Mean) | Adaptiope (Mean) |
|---|---|---|---|
| ST (Zhu, 2005) | 76.6 | 75.4 | 72.7 |
| MOST (GPT-2 (Radford et al., 2019)) | 79.3 | 77.5 | 78.3 |
| MOST (GPT-J-6B (Wang & Komatsuzaki, 2021)) | 79.2 | 77.9 | 78.8 |
| MOST (GPT-3 (Brown et al., 2020)) | 80.1 | 78.9 | 79.9 |

## A.9 MORE DISCUSSION OF MULTI-TEXTUAL PROMPT LEARNING

As described in the main manuscript, the proposed multi-textual prompt learning fuses text prompts in a two-step manner: 1) uniformly average the multiple text features to acquire an initial prompt centroid 2) calculate the final prompt centroid by weighted average where the weight of each feature is the distance between it and the initial prompt centroid. This two-step operation allows smooth prompt fusion by weighting down the effect of corner cases, which is important for multi-textual prompt Learning as the LLM-generated prompts are not always reliable (e.g., when experiencing generation failures, LLM may generate only a full stop character "." or a random word).

In this section, we conduct experiments with ViT-B/16 on ImageNet to investigate the effect of this two-step feature fusion strategy. Table 15 shows that the two-step feature fusion strategy brings about 0.4% performance improvement on ImageNet, largely because it allows smooth prompt fusion by down-weighting the effect of corner cases.

Table 15: Multi-Textual Prompt Learning (MTPL) with and without the two-step feature fusion strategy. Experiments are conducted with ViT-B/16 on ImageNet. The default implementation is highlighted in gray .

| Method | ImageNet |
|---|---|
| CLIP (Radford et al., 2021) | 68.3 |
| MTPL (w/o two-step feature fusion strategy) | 69.4 |
| MTPL (w/ two-step feature fusion strategy) | 69.8 |

Table 16: Results of w/ and w/o prompt engineering with ViT-B/16 on 5 tasks of SUN397, Food101, GTSRB, DTD and UCF101.

| Method | 5-task Mean |
|---|---|
| CLIP w/o Prompt Engineering | 57.5 |
| CLIP w/ Prompt Engineering | 62.0 |
| MUST w/o Prompt Engineering | 63.3 |
| MUST w/ Prompt Engineering | 65.8 |
| MOST w/o Prompt Engineering | **67.7** |

## A.10 PROMPT ENGINEERING

Both CLIP (Radford et al., 2021) and MUST (Li et al., 2022) mitigate the cross-domain text distribution gap by prompt engineering (Radford et al., 2021) and ensembling, e.g., uniform averaging of 80 hand-crafted prompt templates on ImageNet. Although hand-crafted prompt templates bring clear gains, manually designing prompts for each new image recognition task and domain is laborious and time-consuming and degrades the scalability greatly. As Table 16 shows, without any prompt engineering, the proposed MOST still outperforms CLIP and MUST with clear margins, demonstrating its effectiveness and efficiency in handling new visual recognition tasks without prompt engineering. Note we did not include multi-domain datasets (Saenko et al., 2010; Venkateswara et al., 2017; Ringwald & Stiefelhagen, 2021; Peng et al., 2017) in this experiment due to the lack of hand-crafted prompt templates.

## A.11 MORE PARAMETER STUDIES

As described in the main manuscript, our proposed MOST employs the large language model to generate $K$ text prompts for each class for achieving multi-textual prompt learning. We investigate $K$ by varying it from 10 to 25 with a step of 5, as shown in Table 17. It can be seen that varying $K$ does not affect the proposed MOST clearly, demonstrating that our MOST is quite tolerant to the hyper-parameter $K$.

Table 17: Parameter study for the number of text prompts $K$ with ViT-B/16 on Office. The default value is marked in gray .

| Parameter $K$ | 10 | 15 | 20 | 25 |
|---|---|---|---|---|
| Office (Mean) | 79.9 | 80.1 | 80.1 | 80.0 |

As described in the main manuscript, our proposed MOST introduces multi-visual prompt learning that employs the off-the-shelf image augmentation policies in (Cubuk et al., 2020) to generate $K$ image prompts for all images respectively, which are then selectively fused using pseudo class labels to describe each class. We investigate $K$ by varying it from 10 to 25 with a step of 5, as shown in Table 18. It can be seen that varying $K$ does not affect the proposed MOST clearly, demonstrating that our MOST is quite tolerant to the hyper-parameter $K$.

Table 18: Parameter study for the number of text prompts $K$ with ViT-B/16 on dataset Office. The default value is marked in ⬜ gray .

| Parameter $K$ | 10 | 15 | 20 | 25 |
|---|---|---|---|---|
| Office (Mean) | 80.0 | 80.1 | 80.1 | 79.9 |

### A.12 MORE PSEUDO LABEL ACCURACY FIGURES

In Section 4.6 in the main manuscript, we provide the pseudo label accuracy along the unsupervised adaptation process for Office datasets.

In this section, we provide the pseudo label accuracy figures over more datasets, i.e., Office-home, Adaptiope, VisDA, SUN397, Food101, GTSRB, DTD, UCF101, and ImageNet. Fig. 4 shows the pseudo label accuracy along the unsupervised adaptation process with the backbone ViT-B/16. It can be seen that our proposed MOST generates much more accurate pseudo labels than the vanilla self-training (ST) and the state-of-the-art MUST consistently over various datasets. The superior pseudo label accuracy is largely attributed to the proposed multi-prompt denoised learning which helps capture rich target image and text information that is more invariant to visual and textual domain discrepancies and can "prompt" better unsupervised self-training.

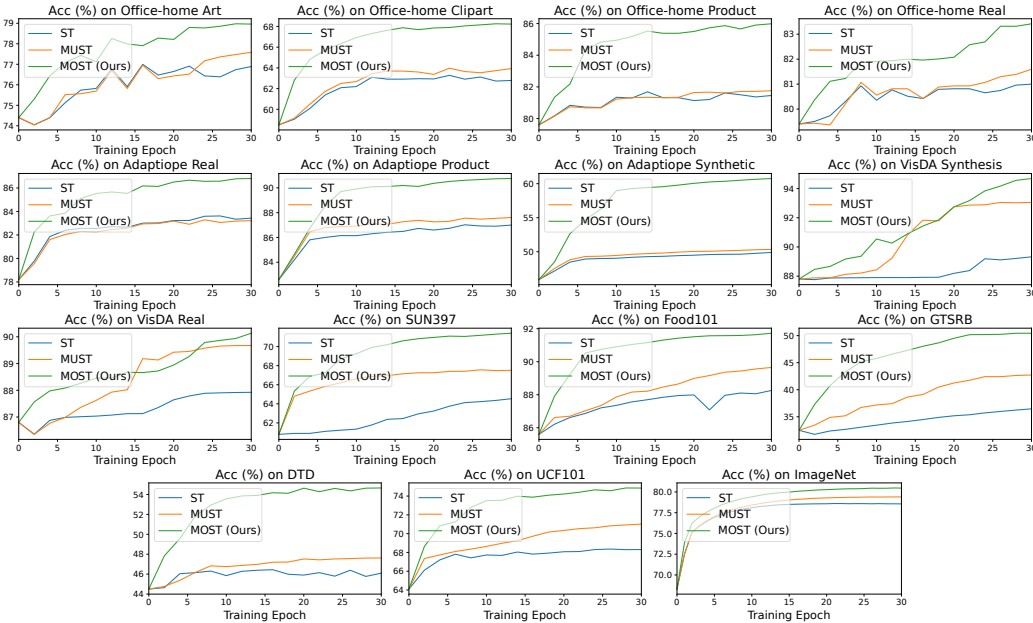

Figure 4: Pseudo label accuracy along the unsupervised adaptation process in OVMA: The experiments were conducted over 10 widely adopted datasets and all use ViT-B/16. The results on dataset Office are provided in the main manuscript.

### A.13 QUALITATIVE RESULTS

We illustrate our proposed MOST qualitatively by providing class activation map (Selvaraju et al., 2017) (CAM) visualization on dataset Office with ViT-B/16. Fig. 5 provides the CAMs of ST (Zhu, 2005) (2nd column), MUST (Li et al., 2022) (3rd column) and our MOST (4th column). We can observe that our proposed MOST preforms image recognition based on more diverse image regions, leading to robust and accurate visual recognition under large cross-domain discrepancies. For example, in the recognition of backpack, MOST tends to rely on more image regions (e.g., various local regions with zippers) which together form a holistic representation of this backpack, ultimately leading to a robust prediction under large domain discrepancies. As a comparison, ST (Zhu, 2005) and MUST (Li et al., 2022) make predictions largely according to a single image region and pay less attentions on other image regions, which may lead to performance degradation when experiencing

large domain discrepancies. The CAMs of Mountain Bike and Helmet shown in the second and third rows respectively are consistent with the above observation.

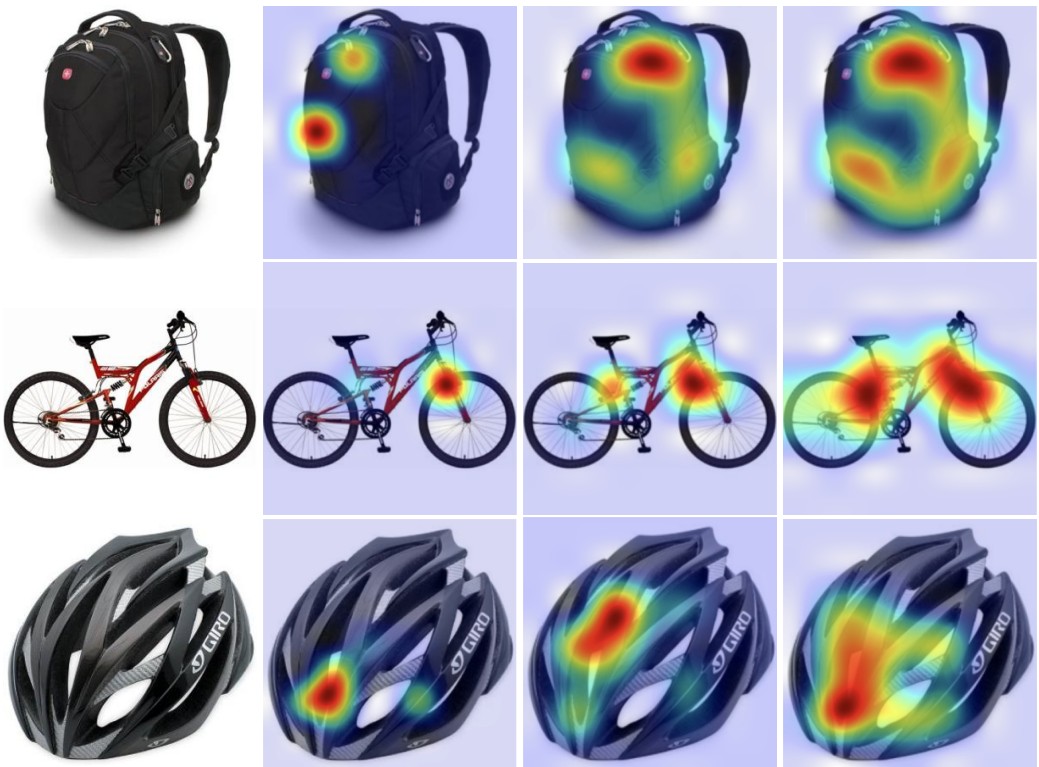

Figure 5: **Qualitative comparisons** with class activation maps (Selvaraju et al., 2017) (CAM) on dataset Office with ViT-B/16. The 4 columns from left to right show Input Images and the corresponding CAMs by ST (Zhu, 2005), MUST (Li et al., 2022) and our MOST, respectively. It can be observed that MOST preforms image recognition based on more diverse image regions, leading to more robust and accurate visual recognition under various cross-domain scenarios.

## A.14   ANALYSIS WITH ERROR BARS

In experiments, we observe negligible variance on the results between multiple random runs. Nevertheless, we provide the error bar with 5 random runs to analyze the proposed MOST with ViT-B/16 on Office dataset, as shown in Table 19. It shows that our proposed MOST performs well consistently over multiple random runs.

Table 19: Analysis of our proposed MOST with error bars. Experiments are conducted with ViT-B/16.

| Method | Office (Mean) | Office-home (Mean) | Adaptiope (Mean) |
|---|---|---|---|
| MOST | $80.1 \pm 0.1$ | $78.9 \pm 0.1$ | $79.9 \pm 0.2$ |

## A.15   BROADER IMPACTS AND LIMITATIONS

We envision that this work will promote more studies on OVMA, a new unsupervised model adaptation framework that mitigates the image annotation constraint and facilitate deep network training while handling new visual recognition tasks. Furthermore, as our work is built upon open-source pre-trained vision-language models, it adds only a small amount of computation overhead after VLM pre-training and therefore reduces the carbon footprint. Currently, we do not foresee clear undesirable impacts of this work from both ethical and social aspects. At the other hand, the investigated techniques in this work are still at a very early stage and thus the proposed approach could be used as an assistant tool

in computer vision applications instead of the critical decision and hard control systems that may lead to severe and harmful consequences.

