# OpenReview forum: "Multi-Prompt Denoised Self-Training for Open-Vocabulary Model Adaptation"
_ICLR.cc/2024/Conference — ICLR 2024 Conference Withdrawn Submission_

### Official Review · Reviewer_Zzqn · 2023-10-30

**Soundness:** 3 good
**Presentation:** 3 good
**Contribution:** 2 fair
**Rating:** 5
**Confidence:** 4

**Summary:**

In this paper, the authors aimed to tackle the problem of model adaptation in the open-vocabulary setting, where the testing labels are not fully covered in training data. Specifically, they leveraged the vision-language model CLIP and designed a new strategy that generates higher-quality pseudo-labels for the model to adapt. Experimental results on multiple benchmarks show the effectiveness of the proposed method.

**Strengths:**

1. The introduction of the open-vocabulary model adaptation framework is interesting and meaningful.
1. The proposed denoised self-training strategy (Eq 7) is novel and interesting.
1. The method shows a significant performance boost.

**Weaknesses:**

1. The authors state that their method “employs a Large Language Model to generate multiple text prompts for a given class name”. Could the authors provide details and perhaps examples of how the multiple-text prompts are generated?
1. It seems that the proposed multi-visual prompts are simply images applied with different augmentation strategies. I have doubts about whether this can be effectively related to prompt learning.
1. The image-text prompt centroid is derived by adding the image prompt centroid to the text prompt centroid. However, it is rather unintuitive to add prompts of different modalities together directly. Is there any support for doing so?
1. “Generalization across different domains and tasks” and “Generalization across different backbones” of Section 4.4 are redundant. The materials are already covered in Sections 4.1 and 4.2

**Questions:**

Overall, while the framework is interesting, using CLIP to tackle open-vocabulary problems is not new. Furthermore, the method lacks a more rigorous explanation, e.g., multi-visual prompts are unclear and directly adding image and textual features is unjustified. It would be very helpful if the authors could respond to the weaknesses I raised above.

---

### Official Review · Reviewer_5Vkc · 2023-10-31

**Soundness:** 3 good
**Presentation:** 3 good
**Contribution:** 3 good
**Rating:** 5
**Confidence:** 5

**Summary:**

The paper presents an exploration of new-task - open-vocabulary model adaptation, detailing a simple yet effective approach to adjust a pre-trained VLM to an unlabeled target set characterized by distinctively varied images and vocabularies. The study delves into three primary relationships for prompt learning, encompassing visual, textual, and temporal (with EMA updates) cues. By employing these prompts, there's a notable stabilization in self-training within the target domain. Notably, the pseudo-label accuracy derived from this approach surpasses that of earlier baseline methods.

**Strengths:**

+ **Presentation quality and clarity**: The manuscript's clarity and organization stand out, enabling readers to trace the core contributions easily. The notations are used properly and the presented pseudo-code algorithms further help understand the proposed approach.
+ **Significance on experimental results**:  The experimental comparison is commendably exhaustive and the proposed method achieves a significant improvement. The authors have put forth comparisons across 11 public benchmark datasets, encompassing both unsupervised domain adaptation and few-shot adaptation protocols. Furthermore, the juxtaposition of the CLIP, CoOp, and CoCoOp tuning methodologies with the paper's proposal is illuminating.

**Weaknesses:**

- **Originality**: The paper's primary technical innovation seems somewhat confined. At its heart, the approach integrates visual (through augmentation) and textual prompts (through GPT3) with the EMA strategy to enhance self-training. Those modules are widely used in prompt tuning, which might limit the novelty of this work.
- **Literature Review**: The literature review appears to be incomplete. Even though the task name is new, the study's main idea is similar to prompt tuning used for domain adaptations. The absence of discussions around pivotal works, especially DAPL [1], is a noticeable gap. Authors are highly recommended to add sections and highlight the differences or connections to this work.
- **Baselines**: Although this work has incorporated numerous experimental comparisons, a very important baseline is missing - leveraging CLIP’s outcomes for pseudo-labelling since its superior efficacy shown in the ablation study section. Given that the proposed method draws on the pre-trained text encoder, the pre-training data potentially introduces additional knowledge. It's pivotal to discern whether the observed performance enhancement stems from this extra data or the inherent qualities of the prompt ensemble. This necessitates examining a baseline that integrates CLIP’s outputs within the self-training loop.
- **Baseline Results**: There are inconsistencies observed in the results presented for the baseline CLIP across Table 1 and Table 3 when compared to the findings documented in references [1,2]. Although both these cited sources are preprints from arXiv, addressing these discrepancies would bolster the paper's credibility and alleviate potential confusion surrounding the method's efficacy.


- **Minor Comment**: The title might not fully reflect the core focus of the paper. Using 'denoising' in the title could be a bit unclear in relation to the main content. Would it be possible to clarify this connection?
----
[1] Ge, Chunjiang, et al. "Domain adaptation via prompt learning." arXiv preprint arXiv:2202.06687 (2022).
[2] Chen, Haoran, et al. "Multi-prompt alignment for multi-source unsupervised domain adaptation." arXiv preprint arXiv:2209.15210 (2022).

**Questions:**

The aspects I'd appreciate further clarification on include:

1. The root cause behind the performance enhancement observed. Is the boost primarily due to the integration of extra data coupled with pseudo labeling? The significance of a baseline, specifically CLIP+PL, in shedding light on this matter cannot be overstated.
2. I'm keen to understand the nexus between the proposed open-vocabulary learning for domain adaptation and prompt tuning for domain adaptation. Grasping this connection would greatly aid in contextualizing the presented findings.

If the authors can provide comprehensive responses to these queries, I would be inclined to elevate my rating.

---

### Official Review · Reviewer_yo5V · 2023-10-31

**Soundness:** 3 good
**Presentation:** 3 good
**Contribution:** 3 good
**Rating:** 6
**Confidence:** 3

**Summary:**

This paper proposes MOST, an open-vocabulary source-free domain adaptation method.
The key to the proposed method is to improve the accuracy of pseudo labeling to the target data by using an ensemble of multiple image and text prompts (average and EMA).
Experimental results on 11 different datasets show that the proposed method gives superior performance under various conditions compared to existing source-free domain adaptation methods.

**Strengths:**

* This paper addresses open-vocabulary source-free domain adaptation based on vision-language models, which is the task highly practical.

* The idea of increasing the accuracy of pseudo labeling by ensembling multiple prompts is intuitive and simple.

* The proposed method outperforms existing popular source-free domain adaptation methods on a wide variety of datasets and evaluation scenarios.

* Exhaustive analyses are reported, including ablation studies, parameter studies, and the cases with different backbones.


* The paper is generally well written.

**Weaknesses:**

a. As the authors also note in this paper, several previous papers have attempted to increase classification accuracy by using ensembles of multiple prompts, so the innovativeness of this paper in terms of technical elements seems somewhat limited.



b. Since VLM is more robust to domain shift than traditional models, the task of adaptation from a fewer number of unlabeled target samples, e.g., test-time adaptation, has attracted much attention. To my understanding, the models are trained on full target data in the experiments conducted in this paper, but it would be nice to have experimental results under conditions where only a few unlabeled target samples are available.



c. Table 17 (or 18) reports the change in performance when the number of text prompts K is varied from 10 to 25, but it would be helpful to include the case of K=1 to make it easier to understand the advantage of using multiple prompts itself.


d. Regarding the process of generating multiple text prompts, it says at the end of P4 that K text prompts were obtained by inputting a class name into GPT-3 (Brown et al., 2020), but it would be better to clarify whether only the class name was input or some richer prompt was used. Also, it would be good to specify the text prompts that were finally generated (= used in the experiment) for reproducibility reason. If you plan to publish the code, it could be included in the code, not in the paper.


e. A.11 has duplicate paragraphs with the same meaning and experimental results (Tables 17 & 18) that should be corrected.

**Questions:**

If I have misunderstood something, I would be grateful if the authors could point it out to me.


I would ask that any innovative technical elements be highlighted in the proposed method as compared to existing methods that use an ensemble of multiple prompts.


I would be grateful if the authors could answer any other questions I have raised as weaknesses.

---

### Official Review · Reviewer_vh13 · 2023-10-31

**Soundness:** 2 fair
**Presentation:** 2 fair
**Contribution:** 2 fair
**Rating:** 5
**Confidence:** 5

**Summary:**

The authors address the issue of open vocabulary domain adaptation, which involves transferring a pre-trained VLM to an unlabeled target domain. VLMs are models that have high generalization capabilities, but sometimes we need them to specialize in a specific field. Finding a way to convert VLMs to be more domain-specific through unsupervised means is a valuable research endeavor. To tackle this problem, the authors propose using multiple text and image prompts for target domain adaptation. Additionally, they update the image encoder using multi-temporal prompt learning, which is a gradually temporal updating method. The experimental results demonstrate the effectiveness of this approach.

My main concerns are regarding the technical novelty and the unclear experimental settings:

The technical novelty of the proposed method is limited. The use of prompts for domain adaptation has been explored in previous works [1]. Furthermore, the concept of temporal prompt learning appears to be a general exponential moving average technique used in semi-supervised learning.

Some of the experimental settings should be explained more clearly. For example, ST (Zhu, 2005) is a survey of semi-supervised learning. What is the exact method being referred to here? Additionally, SHOT is mentioned as addressing source-free domain adaptation. The paper should provide a fair comparison by demonstrating whether the backbone of SHOT is derived from CLIP or a source-pretrained model. If the backbone is from a source-pretrained model, it is important to explain how fairness is ensured in the comparison.

In summary, the authors tackle an interesting problem in their research. However, the paper's writing could be more rigorous before publication.


[1] Domain Adaptation via Prompt Learning.

**Strengths:**

1. The research problem is intriguing and holds significant meaning.
2. The technical solution is clear and the method is easy to replicate.

**Weaknesses:**

1. The technical novelty may be limited.
2. Some experimental settings shoud be explained more clearly.

**Questions:**

See above.